# Vitamin D and SARS-CoV2 infection, severity and mortality: A systematic review and meta-analysis

Oriana D'Ecclesiis[1‡], Costanza Gavioli[2‡], Chiara Martinoli[1], Sara Raimondi[1], Susanna Chiocca[1], Claudia Miccolo[1], Paolo Bossi[3], Diego Cortinovis[4], Ferdinando Chiaradonna[5], Roberta Palorini[5], Federica Faciotti[1], Federica Bellerba[1], Stefania Canova[4], Costantino Jemos[6], Emanuela Omodeo Salé[6], Aurora Gaeta[1], Barbara Zerbato[5], Patrizia Gnagnarella[2]*, Sara Gandini[1]*

1 Department of Experimental Oncology, IEO, European Institute of Oncology IRCCS, Milan, Italy, 2 Division of Epidemiology and Biostatistics, IEO European Institute of Oncology IRCCS, Milan, Italy, 3 Medical Oncology, Department of Medical and Surgical Specialties, Radiological Sciences and Public Health University of Brescia, ASST-Spedali Civili, Brescia, Italy, 4 SC Oncologia Medica, Asst H S Gerardo Monza, Monza, Italy, 5 Department of Biotechnology and Biosciences, University of Milano-Bicocca, Milan, Italy, 6 Department of Clinical Pharmacy, IEO, European Institute of Oncology IRCCS, Milan, Italy

‡ OD and CG are co-first authors on this work.
* patrizia.gnagnarella@ieo.it (PG); sara.gandini@ieo.it (SG)

**Data Availability Statement:** All relevant data are within the paper and its Supporting Information files.

## Abstract

To assess the evidence on SARS-CoV2 infection and Covid-19 in relation to deficiency and supplementation of vitamin D, we conducted a systematic review up to April 2021. We summarised data from 38 eligible studies, which presented risk estimates for at least one endpoint, including two RCT and 27 cohort-studies: 205565 patients with information on 25OHD status and 2022 taking vitamin D supplementation with a total of 1197 admitted to the ICU or who needed invasive mechanical ventilation or intubation and hospital stay, and more than 910 Covid-19 deaths. Primary outcomes were severity and mortality and the main aim was to evaluate the association with vitamin D supplementation. Random effects models showed that supplementation was associated with a significant lower risk of both Covid-19 severe disease (SRR 0.38, 95% CI 0.20–0.72, 6 studies) and mortality (SRR 0.35, 95% CI 0.17–0.70, 8 studies). There were no statistically significant dose differences between studies: summary estimates with regular doses remain statistically significant, suggesting that higher doses are not necessary. For patients on vitamin D supplementation, a greater reduction in mortality risk emerged in older individuals and at higher latitudes. Regarding the quality of studies, assessed using the New Castle-Ottawa quality scale, the analysis revealed in most cases no statistically significant differences between low, medium or high quality studies. We found significant associations of vitamin D supplementation with Covid-19, encompassing risks of disease worsening and mortality, especially in seasons characterized by 25OHD deficiency and with not severe patients. Dedicated randomized clinical studies are encouraged to confirm these results.

**Funding:** Grant Regione Lombardia. (DECRETO N. 7082 Del 17/06/2020. Identificativo Atto n. 366: COVitaminD Trial: prevenzione di complicanze da COVID-19 in pazienti oncologici). The European Institute of Oncology, Milan, Italy is partially supported by the Italian Ministry of Health with Ricerca Corrente and 5×1,000 funds.Federica Bellerba is a Ph.D. student within the European School of Molecular Medicine (SEMM). The funders had no role in study design, data collection and analysis, decision to publish, or preparation of the manuscript.

**Competing interests:** The authors have declared that no competing interests exist.

# Introduction

The pandemic of the novel severe acute respiratory syndrome coronavirus 2 (SARS-CoV2) is a global health threat, with 172,630,637 confirmed cases and 3,178,683 deaths from Covid-19 recorded as of June 6th 2021 [1]. The pandemic had a terrible impact on all societies [2], especially among the elderly. Covid-19 results in a broad spectrum of disease, with great differences in number of cases, severity and fatality rates across countries or regions [3] including more than 80% of patients showing few or no symptoms. Vitamin D (VD) can derive from endogenous or exogenous sources. Endogenous VD is produced in the skin upon exposure to sunlight, due to the capacity of UVB radiation (wavelength from 290 to 315 nm) to convert 7-dehydrocholesterol to previtamin D3, then converted to vitamin D3 [4, 5]. Vitamin synthesis in the skin depends on sunlight exposure, skin pigmentation, age and latitude, especially in Caucasians [6, 7]. Exogenous VD derives from different sources with consumption varying across different regions, dietary pattern and countries. However, aside dietary supplements, the richest foods in VD are fish and fish oil, egg yolks, and to a lesser extent mushrooms. Due to low dietary intake and limited sun exposure, it is often difficult to reach adequate plasma levels of this hormone. Indeed, vitamin D deficiency (VDD) is a major global health problem especially among pregnant women, dark-skinned individuals, obese children and adults. Darker skin and obesity independently increase VDD risk [8–10].

Beneficial effects of adequate VD levels have been shown in different fields, including all-cause mortality [11], seasonal influenza [12], increases susceptibility to respiratory tract infections [13–15], particularly of the upper respiratory tract, and acute respiratory distress syndrome [16, 17].

It has recently been hypothesized that VD status may also influence the severity of responses to Covid-19, and VDD can increase hospitalization and mortality [18, 19]. However, studies analyzing the use of VD supplementation to modify Covid-19 outcomes have offered inconclusive results: while some observational studies in hospitalized patients have shown reduced Covid-19 severity or mortality in patients supplemented with cholecalciferol or calcifediol [20–22], another study described a trend to an increased mortality in patients supplemented with calcifediol [23]. The effects of VD supplementation as a treatment for hospitalized Covid-19 patients have also been studied in three low-powered clinical trials, but no significant reduction in Covid-19 mortality was observed [24–26].

The possible association between Covid-19 and VD could be related to VD ability to inhibit inflammatory response and increase innate defence mechanisms against pathogens [27]. In fact, the biologically most active VD metabolite, 1,25-dihydroxyvitamin D (1,25(OH)2D) is able to repress nuclear factor kappaB (NF-κB)-dependent pro-inflammatory cytokine secretion, such as IL-6 and IL-2, by blocking NF-κB p65 activation via up-regulation of the NF-κB inhibitory protein IkB-α [28]. Furthermore, VD increases secretion of interferon γ (IF γ) and tumor necrosis factor α (TNFα) and regulates the immune system through inhibition of T helper cell type 1 response and stimulating T cells [29]. Moreover, human airway epithelial cells constitutively expressing the vitamin D receptor (VDR) and secretingVD, may lead to increased expression of antimicrobial peptides (such as cathelicidin and defensin beta 4), in adjacent macrophages and other innate immune cells improving anti-viral defenses [30].

Interestingly, hepcidin (HAMP), a protein involved in the controls of the levels and distribution of iron, is upregulated by viral infection causing poor outcomes [31, 32]. Such increase, observed in liver cells and in human peripheral blood mononuclear cells is significantly repressed by 1,25-dihydroxyvitamin D treatment, suggesting another vitamin D3 dependent mechanism in immune function [31].

Moreover, the association between VD and Covid-19 may depend on angiotensin converting enzyme 2 (ACE2), the main host cell receptor for SARS-CoV2 [33] and a fundamental component of the renin-angiotensin-aldosterone system (RAAS) [34], the latter relevant for pulmonary, cardiovascular and kidney function. Indeed, ACE 2 is expressed in organs' tissues including lung, kidney, cardiomyocytes, cardiac fibroblasts, nasal, vascular smooth muscle and endothelial cells. Viral entry prompts both quick viral replication and a down-regulation of ACE2 receptors. The reduction of ACE2 receptor causes an increased production of angiotensin II [35], which is associated to cardiopulmonary and kidney injury as well as ARDS (acute respiratory distress syndrome), all occurring in severe Covid-19 cases [36, 37]. Importantly, VD is a potent inhibitor of renin, the proteolytic enzyme involved in angiotensin I and II accumulation, highlighting the possibility of a VD-dependent negative effect on Covid-19 severity [35].

Finally, it was suggested that VD agonists, such as calcitrol, exhibit protective effects against acute lung injury by modulating the expression of members of ACE2 in lung tissue, and thus supporting the role of VDD as a pathogenic factor in Covid-19.

We performed an updated comprehensive review and meta-analysis covering the current evidence on Covid-19 and VD, in term of SARS-CoV2 infection and disease severity and mortality.

## Materials and methods

The protocol of this systematic review and meta-analysis is available online on PROSPERO; registration number CRD42021255023.

### Sources of information and search strategies

A systematic literature search was conducted and reported following the meta-analysis according to the PRISMA guidelines (S1 Table in S1 File) [38]. Published reports were gathered from the following databases: PUBMED, Ovid Medline, EMBASE, and ISI Web of Science up to April 2021 and conducted by 2 independent reviewers (CG and SR). The publications were retrieved using search terms and text words: "Vitamin D" or "25-OH-D" or "25-hydroxychole-calciferol" or "25-hydroxyvitamin D" or "25-hydroxyvitamin D3" or "cholecalciferol" or "calcitriol" or "hydroxycholecalciferols" in combination with "Covid" or "SARS-CoV2" or "Covid-19" or "Covid-19 mortality" or "Covid-19 death" or "Covid-19 severity" or "SARS-CoV2 infection" without any restriction. The database search was supplemented by consulting the bibliography of the articles, reviews and published meta-analysis.

### Eligibility criteria

We considered original research articles, either experimental, observational or clinical trials. No language or pre-print restriction was applied.

The studies that met the PICO criteria were included (S2 Table in S1 File).

The following inclusion criteria were used for study selection: 1) Research involved individuals tested for SARS-CoV2, or Covid-19 positive; 2) Serum VD levels were available and tested at baseline or within one year from study start, or information was available on baseline VDD, or VD regular use or supplementation for interventional studies; 3) Sufficient information was available to estimate the risk estimate and 95% confidence intervals (CIs) of SARS-CoV2 infection or Covid-19 progression; 4) Studies had to be independent and not duplicate results published in other articles.

We excluded ecological studies, as they did not measure VD levels in the population, studies with not enough data to estimate relative risk, hazard ratio or odds ratio, articles not related to

human viruses/infections or focused on other respiratory chronic diseases, trial protocols not reporting study results and studies presenting only data indicating the increased risk per one unit of 25(OH)D.

## Study selection, data extraction and evaluation of the methodological quality of the studies included

For eligible studies, a standardized data-collection protocol was used to gather the relevant data from each selected article. Two authors (SR and CG) identified the studies independently based on title and abstract, followed by full-text reading and evaluation of study quality. Any disagreement was resolved by consensus.

Data extraction was performed by 3 researchers (PG, CM, CG) using two electronic spread-sheet, one reporting data on the relationship between VD and SARS-CoV2 infection, and the other one for VD and Covid-19 severity.

For the analysis of VD and SARS-CoV2 infection, the following information were collected: first author, year of publication, accrual period, country, study design, source of cases/controls, sample size, age (mean and standard deviation), 25(OH)D levels, number of infected subjects, risk estimate and the corresponding CI, along with possible confounders and study outcome.

For the analysis of VD and the course of Covid-19 we reported the following information: first author, year of publication, accrual period, country, study design, methods for Covid-19 diagnosis, source of the cohort, sample size, age (mean and standard deviation), outcomes (admission to intensive care unit; death), 25(OH)D levels, VD supplementation in case of interventional studies, risk estimate and the corresponding CI, along with possible confounders.

## Study outcomes and exposure

We considered two primary outcomes: Covid-19 severity, in terms of need for admission to hospital intensive care unit (ICU) or for invasive mechanical ventilation or intubation and hospital length of stay (1) or mortality (2). In addition, we considered as secondary outcome SARS-CoV2 infection (3).

We considered two main factors linked to VD exposure: (1) baseline VD deficiency or insufficiency (analysed as lowest versus highest 25(OH)D levels, the most studies used thresh-olds equal to 20nmol / L or 30 nmol / L), and (2) VD supplementation.

## Quality assessment

We used the New Castle-Ottawa quality scale to assess the quality of observational studies and the Cochrane Risk of bias tool to assess the quality of clinical studies included in the analysis (S3-S6 Tables in S1 File).

## Statistical analysis

Depending on the studies, we used odds ratios (ORs), hazard ratios (HRs) and relative risk (RRs) as comparable estimates of the RR.

Random effect models were used to calculate summary relative risk (SRRs) and 95% CIs to evaluate the association between lowest versus highest level of serum VD, or vitamin D supple-mentation versus no supplementation, and SARS-CoV2 infection or Covid-9 severity or Covid-19 mortality. In total, five meta-analyses were performed, according to combinations of the different outcomes and exposure of interest.

Statistical heterogeneity was evaluated through the $I^2$ index and the $X^2$ test. In particular, $I^2 < 50\%$ was considered as indicator of not statistically significant between-study heterogeneity.

Rainforest plots were used as an alternative to conventional forest plots. In this graph type, a likelihood curve for a chosen confidence interval (e.g., 95%) is plotted and then mirrored, producing a shape reminiscent of a raindrop [39].

Subgroups analyses and meta-regression were used to investigate sources of between-study heterogeneity, in terms of study design, enrollment period (especially focusing on the beginning of pandemic period and on the season), geographic area, type of severity, quality score (assigned to each study by the NewCastle Ottawa or Cochrane tool [40]) and adjustment of risk estimates for confounders (e.g. gender, age, comorbidities, body mass index (BMI)). In order to investigate the influence of different doses, we calculated the average IU per month in each study. For studies reporting the information, meta-regression analyses were conducted to assess the potential contribution of latitude and mean age on SRRs. In addition, sensitivity analyses were carried out to investigate the stability of the pooled estimates with respect to each study by excluding individual studies from the analysis. Possible publication bias was evaluated with Egger's linear regression and Begg's correlation test. Furthermore, the possible presence of publication bias was assessed through visual inspection of the funnel plot, and exploratory analyzes were performed using trim and/or fill analysis in order to investigate and adjust the SRR estimate. All reported p values were two sided and $p < 0.05$ was considered statistically significant. Meta-analyses were carried out using the Rstudio software (R version 4.0.0).

## Results

### Characteristics of eligible studies

According to the search strategy, we initially identified 495 articles. Full text review was undertaken for 73 potentially suitable articles, 35 studies were excluded because: they contained neither risk estimates nor sufficient data for calculation; they did not measure VD levels; they were not consistent with the objectives of our study; they were not peer-reviewed or due to study design. After application of the inclusion and exclusion criteria (Fig 1), there were 38 independent studies included in the review that were eligible for at least one of the outcome and exposure investigated.

A total of 207587 patients were included in the quantitative analyses. The main characteristics and results of the selected studies are presented in S7-S9 Tables in S1 File. Of the included studies, the majority are retrospective studies (53%) and the most frequent study design is cohort-study (71%).

### Association between baseline serum vitamin D levels and risk of SARS-CoV-2 infection

Eight retrospective studies [20, 41–47] investigated the association between VD levels and SARS-CoV2 infection (Fig 2 and S7 Table in S1 File), of which three are cohort and five are case-control studies. They were performed in Asia (N = 4), in US (N. 2) and Europe (N = 2). Serum VD levels were measured at admission to hospital in case-control studies, or previously (from one year to 14 days before infection). All the studies showed an increased risk of Covid-19 positive test in subjects with lower 25(OH)D levels (Fig 1), and the SRR indicated a significant double increased risk of infection for subjects with low serum VD levels compared to the highest level: SRR = 2.18 (95% CI: 1.55–3.06).

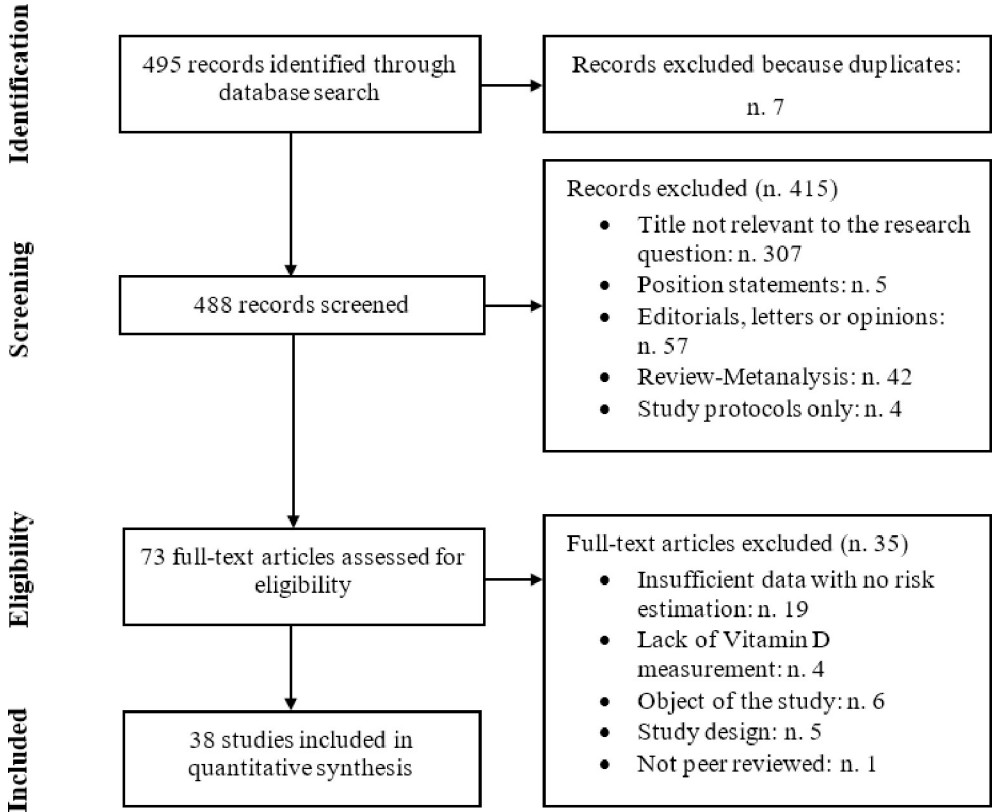

**Fig 1. Flow chart.** Study flow-chart for the process of selecting the enrolled studies.

The eight studies included reported statistical significant increased risk, except the study published by Ferrari [47]. They showed significant heterogeneity ($I^2$ = 87%, p <0.0001). Sensitivity analysis showed that with the exclusion of the study by Hernandez et al. [20], the heterogeneity across studies was strongly reduced and low VD levels remained significantly associated with increased risk of infection (SRR = 1.74, 95% CI 1.44–2.11; $I^2$ = 49%). Instead, when each of the other studies was excluded, heterogeneity remained high. Excluding the study of Kaufman et al. [46], as the only one that reported the values of VD collected before the beginning of the study, results did not change and a significant positive association was confirmed (SRR = 2.33, 95% CI 1.59–3.42).

Stratifying by study design, low VD levels were found to be significantly associated with infection both in cohort studies (SRR = 1.54, 95% CI 1.49–1.59; $I^2$ = 0%) and in case-control studies (SRR = 2.78, 95% CI: 1.71–4.52; $I^2$ = 74%), with stronger estimates in the latter subgroup but higher heterogeneity (p <0.0001 by study design). Considering that all the included studies were retrospective, and most included studies reported unadjusted estimates, it was not possible to evaluate these characteristics by subgroup analyses.

There was no statistically significant difference between the studies conducted in Asia vs Europe and USA (p = 0.73, S10 Table in S1 File).

The study enrollment time period was not found to be significantly associated with between-study heterogeneity. Considering March as enrollment time period or not, we found a significant positive association (SRR = 2.37, 95% CI: 1.13–4.98; $I^2$ = 93%; SRR = 2.02, 95% CI: 1.41–2.89; $I^2$ = 61% respectively, p = 0.72 between the two groups). Finally, applying meta-regression analysis, it emerged that latitude and age were not significantly associated

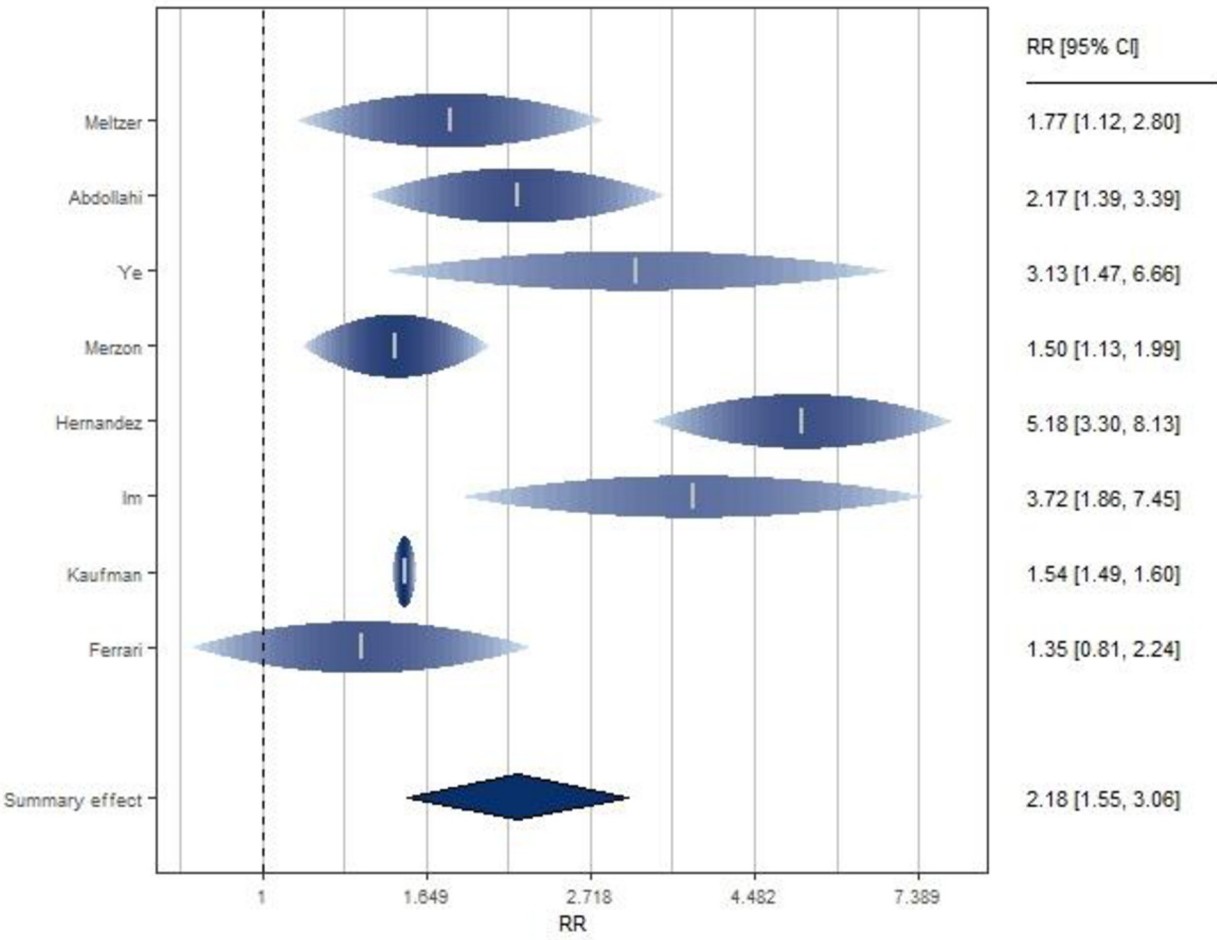

**Fig 2. Forest plot of risk estimates of infection for low vs high vitamin D levels.** Forest plot: Ticks represent effect estimates of individual studies with their 95% confidence intervals. Raindrops and shading represent the probability density for a probability of 0.95 with height of raindrop and color saturation proportional to the weight assigned to the study in the meta-analysis. The diamond represents the overall result and 95% confidence interval of the fixed-effect meta-analysis.

with study estimates (p = 0.84; p = 0.49, respectively) while the score, assigned to each study included to assess their quality, was significantly associated (p = 0.03). Despite this, there was no statistically significant difference between low and medium quality studies (p = 0.30) by subgroup analysis (S10 Table in S1 File).

A dose-response analysis showed that cut-offs of different VD levels did not change the effect (p = 0.90).

No evidence of publication bias was found. Both Egger (p = 0.17) and Begg's test (p = 0.27) supported the null hypothesis of absence of publication bias. Using the trim and/or fill analysis (S1 Fig in S1 File), no potentially missing studies were imputed, further supporting the hypothesis of absence of publication bias.

## Association between baseline serum vitamin D levels and severity of Covid-19

Sixteen studies [20, 48–62] investigated the association between VD levels and severity of Covid-19 in terms of patient need for ICU admission or ventilation requirement or intubation (Fig 3 and S8 Table in S1 File). Serum VD levels were measured at admission to hospital.

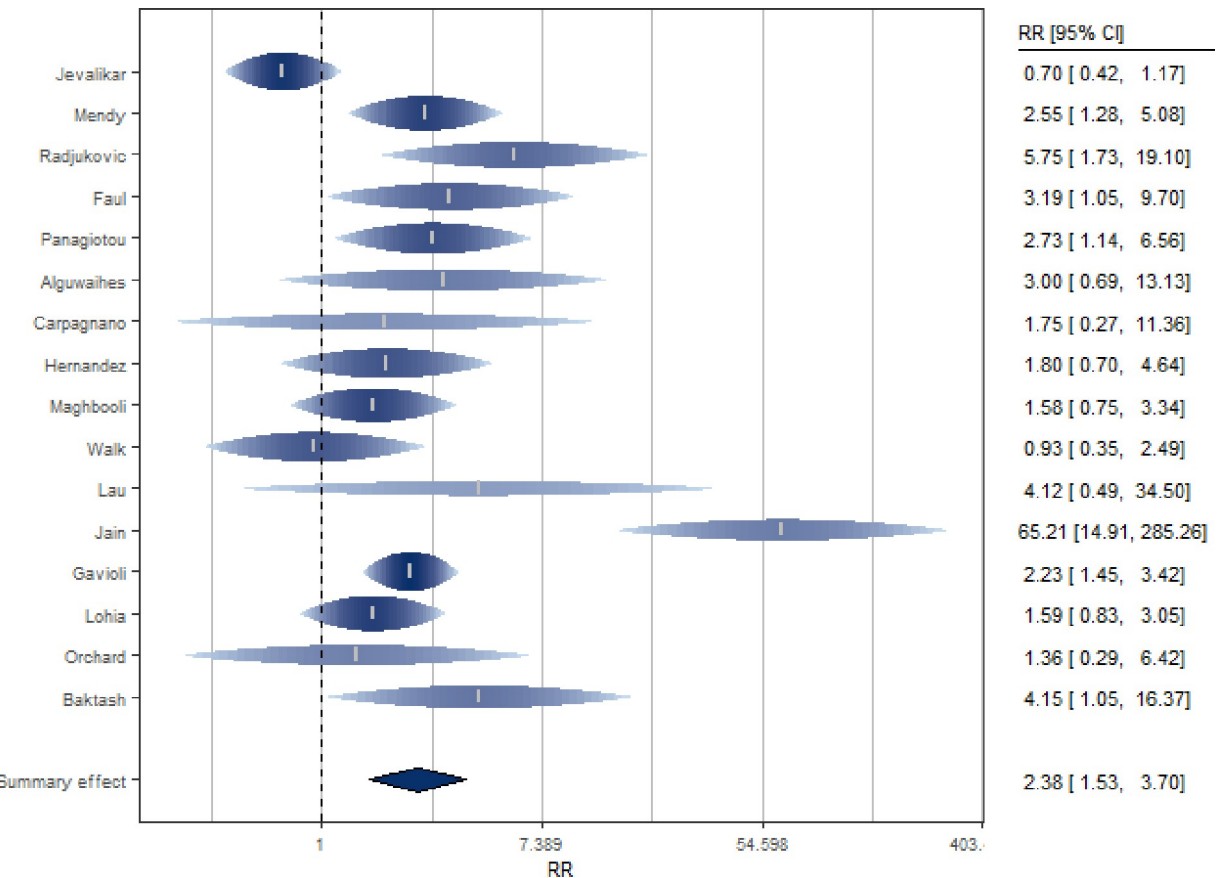

**Fig 3. Forest plot of risk estimates of severity for low vs high vitamin D levels.**

The SRR indicated a significant double increased risk of severity for subjects with low serum 25(OH)D levels (SRR = 2.38, 95% CI: 1.53–3.70) (Fig 3), although significant heterogeneity among studies was detected ($I^2$ = 72%, p <0.0001 for heterogeneity).

Sensitivity analysis indicated that exclusion of the study by Jain et al. [58] strongly reduced the heterogeneity among studies, but poorly affected the summary estimate (SRR = 1.92, 95% CI: 1.39–2.64; $I^2$ = 45%). The sensitivity analysis, conducted excluding from the calculation of the summary estimate the studies [49, 56, 57] that had not yet undergone the peer-review process (pre-print), led to consistent results. Indeed, a significant increased risk with low 25OHD was confirmed (SRR = 2.55, 95% CI: 1.50–4.32).

Through a subgroup analysis, a positive and significant association between low VD levels and Covid-19 severity emerged for the cohort studies (N = 13, SRR = 2.91, 95% CI: 1.79–4.73; $I^2$ = 66%), while a non-significant association was found for the non-cohort studies (N = 3, SRR = 1.16, 95% CI: 0.61–2.19; $I^2$ = 57%). The difference between the two groups was statistically significant (p = 0.03). When only prospective studies (N = 8) were included in the meta-analysis, the association between VD levels and disease severity was confirmed (SRR = 2.59, 95% CI: 1.04–6.46; $I^2$ = 87%), and the difference between prospective and non-prospective study groups was not significant (p = 0.94). There was no statistically significant difference between the studies conducted in Asia, Europe and USA (p = 0.97, S11 Table in S1 File). Furthermore, the association between 25(OH)D levels and risk of severity was observed both in studies that adjusted the estimate for age and sex (N = 5, SRR = 2.29, 95% CI: 1.69–3.09;

I^2 = 0%) and in studies that did not take these two variables into account (N = 11, SRR = 2.40, 95% CI: 1.21–4.76; I^2 = 78%; p for heterogeneity = 0.77), as well as in studies that adjusted the estimate for comorbidities (N = 3, SRR = 2.19, 95% CI: 1.56–3.09; I^2 = 0%) and BMI (N = 3, SRR = 2.06, 95% CI: 1.45–2.92; I^2 = 0%) (S11 Table in S1 File).

The summary estimate remained significant when considering studies that included March 2020 in the enrollment period (N = 3, SRR = 2.15, 95% CI: 1.66–2.79; I^2 = 0%). The estimates lost significance when considering the studies that did not include this month (N = 11, SRR [95% CI]: 4.84[0.35–67.08]; I^2 = 94%). The difference between SRR of these two groups resulted, however, not statistically significant (p = 0.49). There was no significant difference in estimation (p = 0.82) between studies whose data were collected during the summer season and those whose data were collected during the other seasons (N = 4, SRR [95% CI]: 3.53 [0.53–23.48]; N = 12, SRR [95% CI]: 2.22[1.73–2.86], even if a significant association was confirmed only when studies were not conducted in summer. This may suggest that the association with VD may be stronger in winter when deficiency is usually greater. Through a subgroup analysis, no difference emerged between the studies that had as their outcome the admission to ICU compared to the others (p = 0.70, S11 Table in S1 File). Through a meta-regression, it emerged that both latitude and age did not act as risk modifiers (p = 0.82; p = 0.83, respectively). Finally, regarding the quality of the studies, it was found that the score could act as a risk modifier (p = 0.03) and a statistically significant difference between the studies of low and medium quality was observed (p = 0.02), even if the estimates of the latter are both significant and confirm the association (S11 Table in S1 File).

A dose-response analysis showed that cut-offs of different VD levels did not change the effect (p = 0.66).

The methods used to assess the presence of publication bias have led to controversial conclusions. There was no indication of publication bias from Begg's test (p = 0.23) while from Egger's regression (p = 0.04) there was. There is no strong evidence of publication bias from the funnel graph asymmetry check and missing studies were not graphically imputed (S2 Fig in S1 File).

## Association between vitamin D supplementation and the severity of Covid-19

Six studies [20, 48, 63–66] investigated the association between VD supplementation and severity of Covid-19 in terms of need for ICU admission or ventilation requirement or intubation (Fig 4 and S9 Table in S1 File). The studies, mostly prospective, included clinical trials (N = 2), cohort (N = 2), case-control (N = 1) and cross-sectional (N = 1) studies. Most were performed in Europe (N = 3), two in Asia and one in South America.

The SRR for these studies indicated a significant reduction of risk severity (62%) when subjects were supplemented with VD (SRR = 0.38, 95% CI: 0.20–0.72), with no indication of relevant heterogeneity among studies (I^2 = 47%).

In sensitivity analysis, the elimination of one study [24] reduced the heterogeneity to zero while the summary estimate remained significant and indicated a reduced risk of severity for subjects taking VD supplementation (SRR = 0.53, 95% CI: 0.35–0.80). Similarly, excluding Tan's study [65], the results still suggested a 55% risk reduction of severity for subjects supplemented with VD (SRR = 0.45, 95% CI: 0.26–0.80, I^2 = 32%).

Excluding the study administering calcifediol, the estimate is still significant (SRR = 0.56, 95% CI: 0.36–0.85).

There was no statistically significant difference between trials and observational studies (p = 0.61) and between the studies conducted in Europe vs other (p = 0.11) in terms of estimates (S12 Table in S1 File).

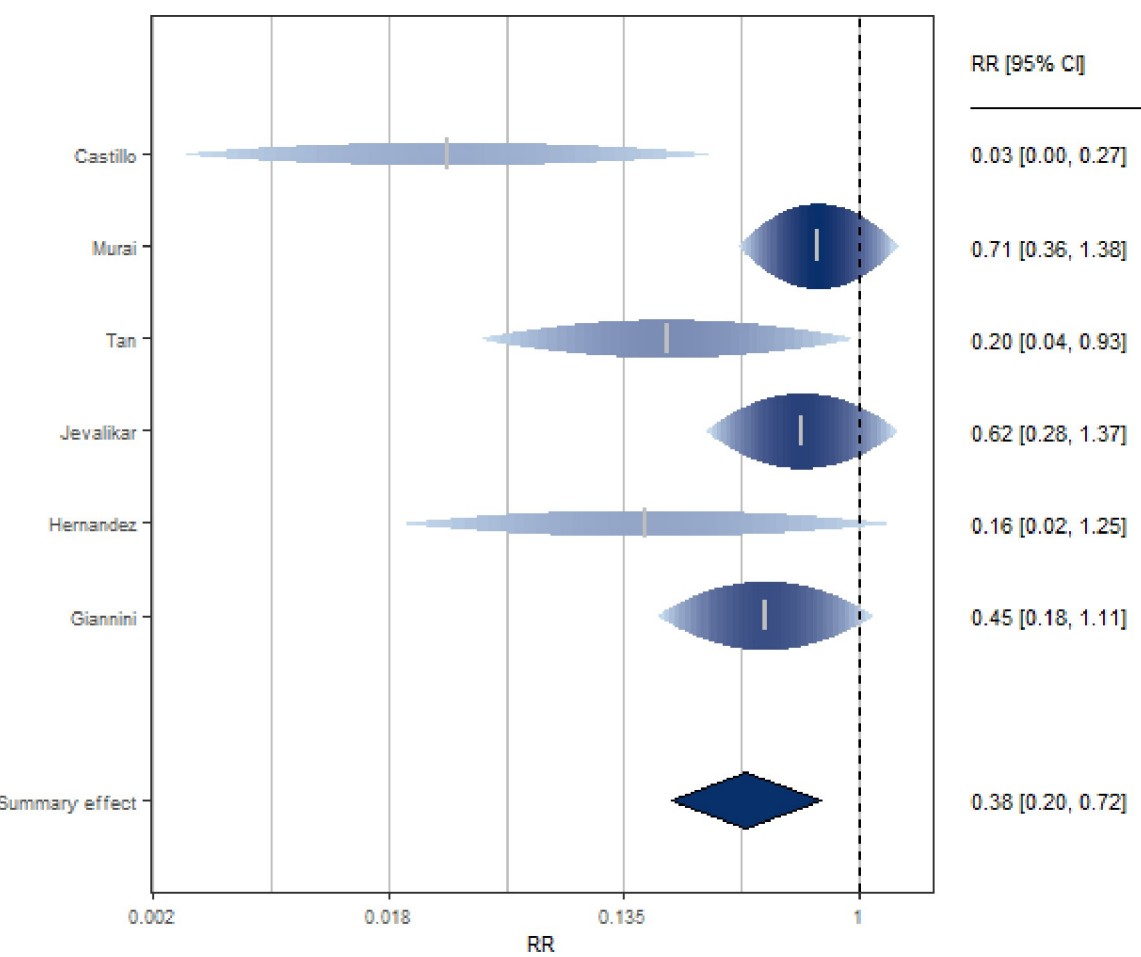

**Fig 4. Forest plot of risk estimates of severity for vitamin D supplementation vs not supplementation.**

When considering studies enrolling patients in March 2020, the estimate was poorly affected (SRR = 0.33, 95% CI: 0.16–0.68;I^2 = 0%). We found that summary estimates from studies whose data were collected during the summer season were significantly greater (p = 0.03) than estimates from studies whose data were collected during other seasons (N = 2, SRR [95% CI]: 0.67 [0.40–1.12]; N = 4, SRR [95% CI]: 0.20 [0.07–0.56], again suggesting that supplementation maybe less important in summer when VDD is less common. Through a subgroup analysis, no difference emerged between the studies that had as their outcome the admission to ICU compared to the others (p = 0.43, S12 Table in S1 File). There was no statistically significant difference between the studies that included an initial bolus and those that did not (p = 0.46, S12 Table in S1 File). In addition, there were no statistically significant differences between the studies involving regular doses and those involving high doses (p = 0.18; p = 0.28, respectively for the ones with an initial bolus and without, S12 Table in S1 File). Further subgroup analyses could not be performed due to the small number of studies.

Through a meta-regression, it emerged that latitude, age and score did not act as risk modifiers (p = 0.96; p = 0.74;p = 0.41, respectively). From the subgroup analysis, there was no statistically significant difference (p = 0.61) between high and medium quality studies (S12 Table in S1 File).

There was indication of publication bias by Egger's test (p = 0.002) and Begg's test (p = 0.016) and checking the asymmetry of the funnel plot (S3 Fig in S1 File). With the analysis of the trim and / or fill, two missing studies were imputed and if we took these into account, the association between VD supplementation and disease severity presented a similar point estimate, but with wider 95%CI; between-study heterogeneity became highly significant (SRR = 0.48, 95% CI: 0.22–1.06; I^2 = 82%).

## Association between baseline serum vitamin D levels and mortality in patients with Covid-19

Nineteen studies [20, 48–50, 53, 54, 58–62, 67–74] investigated the association between baseline VD levels and mortality of Covid-19 patients (Fig 5 and S8 Table in S1 File). Serum VD levels were measured at admission to hospital or previously. The studies, mostly cohort and European studies, include both prospective (N = 10) and retrospective (N = 9) analyses. The SRR for these studies suggested a significantly double increased risk of death for subjects with low level of 25(OH)D (SRR = 2.35, 95% CI: 1.46–3.80), with significant heterogeneity among studies (I^2 = 64%, p for heterogeneity = 0.0002).

After excluding single studies in the analysis, between-study heterogeneity remained statistically significant. By excluding the study by Gavioli et al. [59], the estimate remained significant and suggest more than double increased risk of mortality (SRR = 2.58, 95% CI: 1.56–4.24) with a lower but still significant heterogeneity (I^2 = 56%). The sensitivity analysis conducted excluding from the calculation of the summary estimate the study that had not undergone the peer-review process [49] led to results consistent with the previous ones (SRR = 2.5, 95% CI:

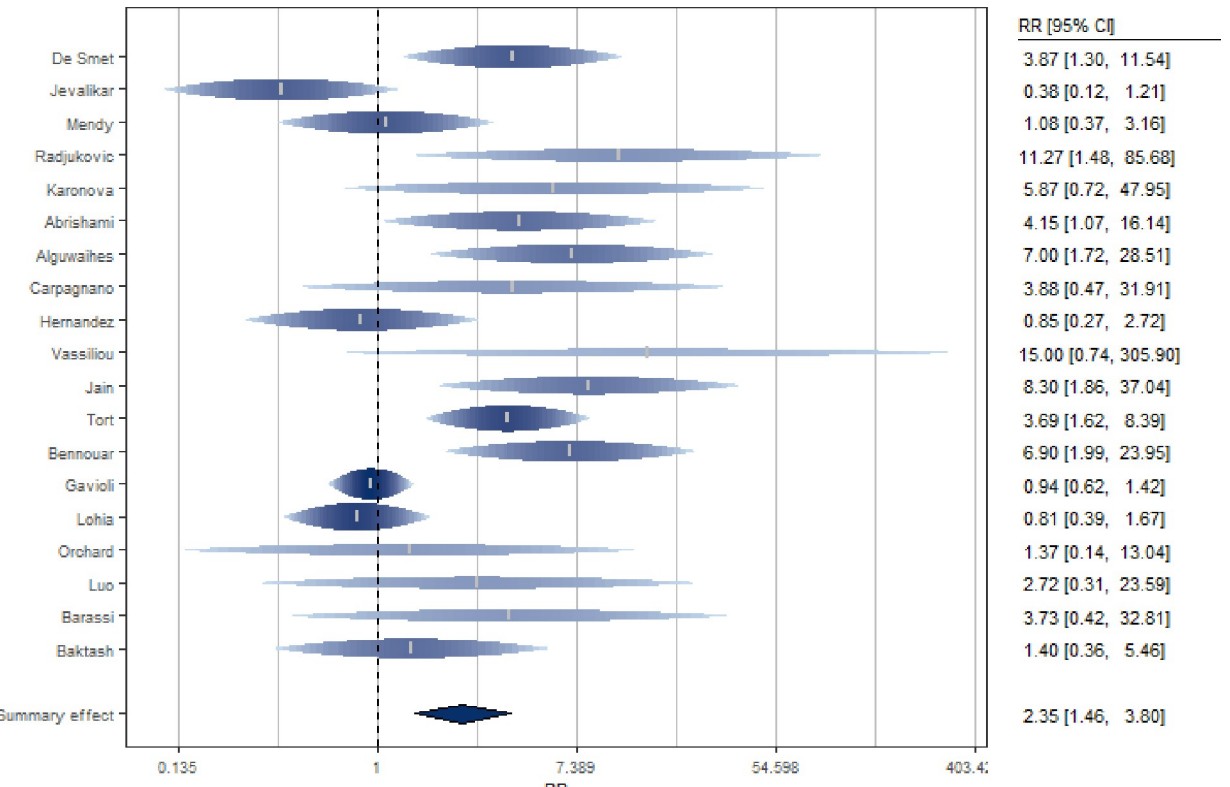

**Fig 5. Forest plot of risk estimates of mortality for low vs high vitamin D levels.**

1.51–4.14). In addition, excluding the study of Orchard et al. [61], as the only one reporting VD values collected before the beginning of the study, a significant positive estimate (SRR = 2.41, 95% CI 1.47–3.95) continues to emerge.

When the meta-analysis was restricted to cohort studies only (N = 15), results suggested an even higher risk of death for subjects with low 25(OH)D levels (SRR = 2.79, 95% CI: 1.63–4.79). Considering only prospective studies (N = 10) led to similar results (SRR = 2.71, 95% CI: 1.34–5.46) with a stronger association compared to the one obtained considering only the retrospective studies (N = 9, SRR = 2.05, 95% CI: 1.05–4.01). There was no statistically significant difference in terms of estimates between the geographical areas (p = 0.18, S13 Table in S1 File). There was no statistically significant difference between the subgroup of studies that included the month of March 2020 in the enrollment period (N = 13, SR = 2.49, 95% CI: 1.39–4.45) and the ones that did not (N = 5, SRR = 2.39, 95% CI: 0.76–7.52), (p = 0.89). There was no statistically significant difference in estimation (p = 0.85) between studies whose data were collected during the summer season and those whose data were collected during the other seasons (N = 6, SRR [95% CI]: 2.88[0.87–9.59]; N = 13, SRR [95% CI]: 2.14[1.32–3.47]. Further, there were no statistically significant differences between estimates adjusted for confounders and crude estimates (S13 Table in S1 File).

Even if latitude (p = 0.69) and age (p = 0.83) were taken into account, a strong residual heterogeneity remained in the model. Finally, regarding the quality of the studies, it was found that the score could act as a risk modifier (p = 0.005) and a statistically significant difference between the studies of low and medium quality was observed (p<0.0001) (S13 Table in S1 File).

A dose-response analysis showed that cut-offs of different VD levels did not change the effect (p = 0.15).

Begg's rank correlation test showed absence of publication bias (p = 0.37) in contrast to Egger's linear regression (p = 0.01). By checking the asymmetry of the funnel plot (S4 Fig in S1 File), there appeared to be evidence of publication bias. Six potentially missing studies were identified which, if available, would have rendered the summary estimate not significant (SRR = 1.55, 95% CI: 0.95–2.53; I^2 = 71%).

## Association between vitamin D supplementation and mortality in patients with Covid-19

Seven studies [20, 21, 48, 63, 64, 75, 76] investigated the association between VD supplementation and mortality of Covid-19 patients (Fig 6 and S9 Table in S1 File). The studies, mostly European, equally distributed between prospective and retrospective, included clinical trials (N = 2), cohort (N = 3), case-control (N = 1) and cross-sectional (N = 1) studies. The SRR suggested a significant 65% reduction in the risk of death for subjects supplemented with VD (SRR = 0.35, 95% CI: 0.17–0.70), with significant heterogeneity among studies (I^2 = 55%).

Through a sensitivity analysis, after exclusion of the study by Murai et al. [64], the heterogeneity between studies became null (I^2 = 0%) an even stronger reduction in the risk of death was observed (SRR = 0.28, 95% CI: 0.18–0.44). Excluding the study administering calcifediol, the estimate is still significant (SRR = 0.32, 95% CI: 0.14–0.69).

There was statistically significant difference between the trials and observational studies (p = 0.01, S14 Table in S1 File). A not statistically significant difference (p = 0.15) emerged between the group of studies that adjusted the estimate for age, gender, comorbidity and "other" with a strong and reduction in risk (SRR = 0.27, 95% CI: 0.16–0.46), compared to the group of studies that did not adjust for the above confounders (SRR = 0.47, 95% CI: 0.15–1.41). This subgroup analysis was repeated including the trials studies in the "adj" group. In

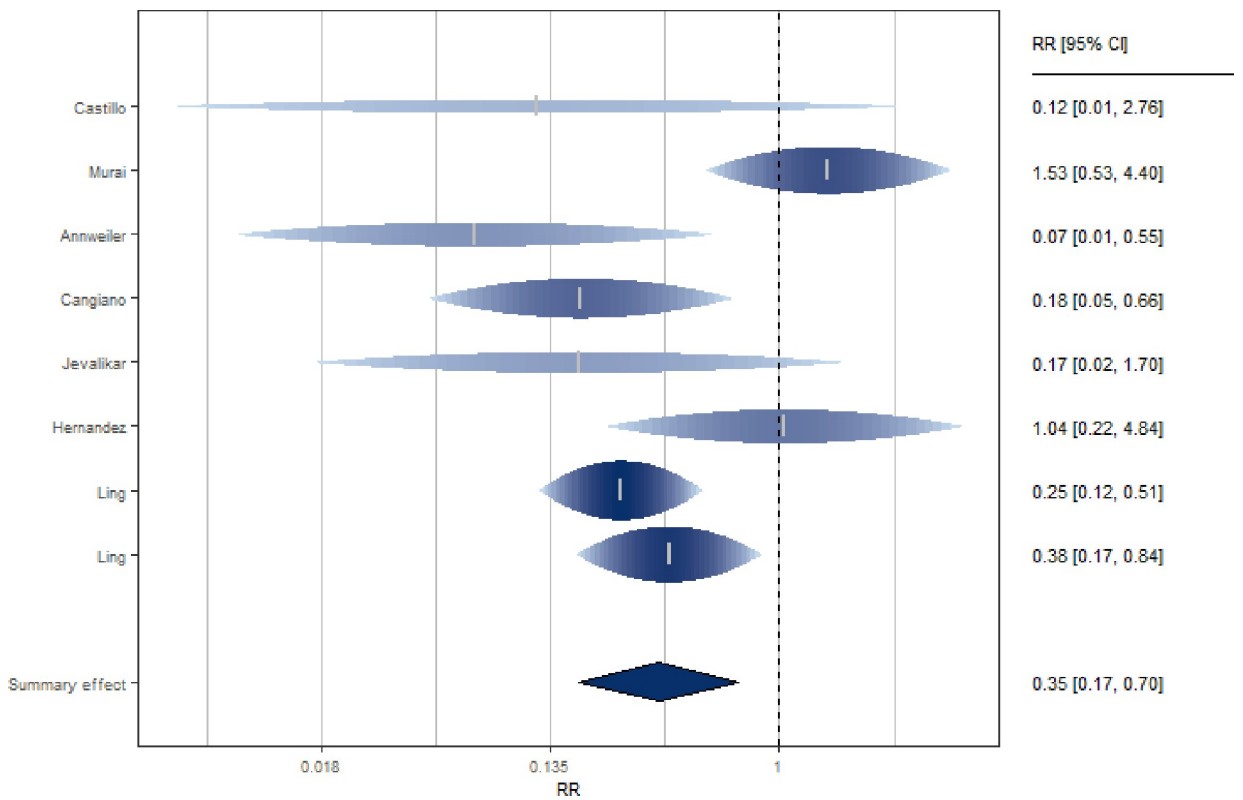

**Fig 6. Forest plot of risk estimates of mortality for vitamin D supplementation vs not supplementation.**

this case the risk estimates confirmed a significant reduction in risk of death in both study groups (SRR = 0.34, 95% CI: 0.13–0.90 for adjusted studies; SRR = 0.33, 95% CI: 0.09–1.16 for 'not adjusted') and the difference adjusted and not adjusted estimates was non-significant (p = 0.89).

There was no significant difference between the prospective and retrospective study groups (p = 0.63; SRR = 0.36, 95% CI: 0.09–1.40 and SRR = 0.31, 95% CI: 0.19–0.51, respectively). When only the cohort studies were considered, the estimate remained statistically significant and suggested a reduced risk with no between-study heterogeneity (SRR = 0.26, 95% CI: 0.16–0.42, $I^2$ = 0%). Considering cohort studies and trials in the same subgroup, the relationship persists (SRR = 0.31, 95% CI: 0.14–0.71; $I^2$ = 64%). There was no statistically significant difference in estimation (p = 0.38) between studies whose data were collected during the summer season and those whose data were collected during the other seasons (N = 4, SRR [95% CI]: 0.44[0.18–1.06]; N = 4, SRR [95% CI]: 0.24[0.07–0.82], even if the association was confirmed to be statistically significant in studies conducted not in summer. There was a significant difference in estimation between the studies conducted in Europe vs others (p = 0.02, S14 Table in S1 File).

There was no statistically significant difference between the studies that included an initial bolus and those that did not (p = 0.79, S14 Table in S1 File). In addition, there were no statistically significant differences between the studies involving regular doses and those involving high doses (p = 0.09; p = 0.09, for the ones with an initial bolus and without, respectively, S14 Table in S1 File).

Both latitude and age acted as moderators of the effect (p = 0.024; p = 0.019, respectively). In particular, a greater reduction in mortality risk was observed for studies with a higher mean age and conducted at high latitudes. Finally, regarding the quality of the studies, it was not found that the score could act as a risk modifier (p = 0.05), but a statistically significant difference between the studies of high and medium quality was observed (p = 0.01), even if the estimates of the latter are both significant and confirm the association (S14 Table in S1 File).

There was no indication of publication bias by Egger's test (p = 0.32) and Begg's test (p = 0.90). Two missing studies were identified, but did not substantially change the results (SRR = 0.42, 95% CI: 0.21–0.84; I^2 = 54%) (S5 Fig in S1 File).

## Discussion

This systematic review and meta-analysis evaluated the relationship between VD (either serum 25(OH)D levels, or VD supplementation) and SARS-CoV2 infection, severity and mortality. We report that VD supplementation was significantly associated with improved Covid-19 outcomes, with more than halved risk of severity and mortality. Furthermore, subjects with low 25(OH)D levels had a significant double increased risk of SARS-CoV2 infection, higher risk of severity and Covid-19 mortality when compared to subjects with higher VD levels. The results of our meta-analyses are in agreement with studies showing that low 25(OH)D levels are associated with increased susceptibility to respiratory infections [16, 77, 78], and that VD supplementation can prevent acute respiratory tract infection [13], and may be explained by the fact that VD can act as a modulator of innate and adaptive immune system responses, through the induction of antimicrobial peptides and the regulation of pro-inflammatory cytokine secretion [79].

Regarding some scenarios, there was significant heterogeneity among studies, however, in some cases, could be reduced in sensitivity analyses.

For example, despite the initial significant heterogeneity detected among studies focused on SARS-Cov2 infection, we found that excluding the study by Hernandez et al. [20] reduced the heterogeneity while maintaining a significant association between serum VD levels and risk of testing positive for coronavirus. The heterogeneity attributed to this study could be explained by the fact that it was the only one collecting data exclusively in March 2020, the initial, crucial month of the emergency. The point estimate of this study is also much higher than the others, indicating an even greater risk of infection for subjects with low VD levels. In line with this reasoning, a subgroup analysis showed that the studies that included March 2020 in the accrual period, presented higher estimates than the studies starting data collection afterwards, although the difference between these groups was not statistically significant. In addition, the study by Hernandez et al. [20], unlike the others, used different methods to assess serum levels in cases and controls.

In the analysis of VD supplementation and the severity of Covid-19 most of heterogeneity could be attributed to the study by Castillo et al. [24]. Notably, this is the only study reporting the adjusted estimate for diabetes and hypertension, two factors associated with VDD and known to increase the risk of Covid-19 severity [80–82]. Similarly, in the meta-analysis on VD supplementation and Covid-19 mortality, it emerged that when excluding the study by Murai et al. [26], the estimate remained significant and the heterogeneity decreased, probably because it is the only study conducted in Brazil, a country that has experienced an acute Covid-19 emergency phase especially since the second half of 2020, while data for this study were collected between June/August 2020 [83].

From our meta-analysis, Covid-19 patients who supplemented their diet with VD had a 73% reduced risk of mortality, taking into account age, sex, comorbidities and other

adjustments. Our findings corroborate the results of a recent meta-analysis of pooled data from RCTs [84] showing that VD supplementation may reduce the risk of acute respiratory infections (ARI) Interestingly, in the studies in which VD was administered daily, a significant protective effect of VD supplementation was observed on the risk of having one or more ARI (OR = 0.78, 95% CI: 0.65–0.94).

The reason why VD supplementation improves severity outcomes for Covid-19 positive people is still unclear, although various mechanisms have been proposed. Mechanistically, the intake of VD could lead to the stabilization of physical barriers, in which cells tightly bound together would prevent foreign agents from reaching the tissues most susceptible to infections [12].

Also, upon analyzing associations of baseline 25(OH)D levels and both severity and mortality, we observed significant heterogeneity among studies. In the meta-analysis focused on Covid-19 severity, heterogeneity was solved through a sensitivity analysis: by excluding the study by Jain et al. [58], the heterogeneity became not significant and the summary estimate remained significant, highlighting a risk relationship between low VD levels and severity in Covid-19 patients. Jain et al. [58] excluded from their analysis, patients with mild and moderate symptoms, which may be the source of the greater heterogeneity between studies.

It should be noted that in the scenario that considers mortality as the outcome, by imputing the potential missing studies (N = 6), the pooled estimate did not become more significant, suggesting the need to further investigate for this outcome.

Blood levels of 25(OH)D are the only reliable indicators of VD status [85]. Several studies show that VD levels are both age and sex sensitive. Some studies suggest that regular VD supplementation has survival benefits in Covid-19 patients [86, 87]. This was particularly evident for the elderly, who probably have a higher beneficial effect of supplementation due to generally lower baseline values than younger individuals and also produce a smaller amount of D3, when exposed to the sun [88], possibly explaining why age was found to be significant in the meta-regression related to VD supplementation and mortality. Regarding association between age, Covid-19 and VD levels, we have also to consider the angiotensin-converting enzyme 2 (ACE2), which acts as a primary receptor for SARS-CoV2 entry into cells and whose expression decreases with age, especially in men. Importantly, the active form of VD increases the expression of ACE2, contrasting the negative effect of SARS-CoV2 virus on ACE2 expression, and therefore underlining a scenario in which elderly patients, supplemented with VD, could benefit from ACE2 protein increased expression [89].

From our results, in the various scenarios, significant associations emerged even when only studies correcting the estimate for possible confounding factors, including sex, were considered, highlighting the reliability of the summary estimates. There are conflicting opinions regarding sex [90]. Thus, it is possible to hypothesize that even in Covid-19, vitamin D3 may play a role in mortality and severity due to its sex-associated immunoregulatory effects [90] and it is thus important to include this variable in the models.

Although there appears to be an association between high latitude and VDD, no negative relationships emerged from our study except for the meta-analysis related to VD supplementation and mortality. One reason may be that those who live in high latitudes are more inclined to supplementation of VD as it is necessary especially during the winter [91]. Estimates from studies conducted in summer are in fact not statistically significant. This may be explained by the fact that in winter and spring VDD is more frequent than in summer.

There were no statistically significant differences in estimates between geographic areas except for the VD supplementation and mortality scenario. It is interesting to note that when heterogeneity is lower, the estimates are consistent and suggest a significant association with VD.

The meta-regressions showed, in most scenarios, that the risk decreases as the quality of the study increases. To further investigate, we conducted subgroup analyzes and assessed whether there were statistically significant differences, in terms of risk estimation, between quality of studies. Statistically significant differences emerged when evaluating associations between VD levels and both severity and mortality and when analyzing the relationships between VD supplementation and mortality. In the first and last cases, the estimates of both subgroups remained significant and confirmed the association.

Dose-response analyses showed no different effects in terms of cut-offs of VD levels in all scenarios. Pereira et al. [92] conducted a systematic review and meta-analysis finding a VDD increased mortality from Covid-19 (SRR = 1.82, 95% CI: 1.06–2.58), in line with our results. Regarding the relationship between low 25(OH)D levels and infection, our results identify a significant positive association unlike those of Pereira et al [92], in which the emerged association was not significant (SRR = 1.35, 95% CI: 0.80–1.88). This study considered low concentrations of VD <50nmol / L, while ours involved studies that reported, for the most part, thresholds equal to 20nmol / L or 30 nmol / L, possibly explaining the discrepancy in the results of the two studies. Our results are consistent with a meta-analysis of observational studies [93] highlighting a significant association between VDD and Covid-19 severity and mortality (SOR = 2.6, 95% CI: 1.84–3.67; SOR = 1.22, 95% CI: 1.04–1.43, respectively) and between low 25(OH)D levels and SARS-Cov2 infection like our study (SOR = 1.26, 95% CI: 1.19–1.34). Compared to this, our is an updated meta-analysis and therefore includes more recent studies. A meta-analysis by Kazemi et. [94] investigated the possible association between low levels of VD and SARS-Cov2 infection, severity and mortality. The results of the latter are in line with ours. The two works differ in terms of methodology, partially different exclusion criteria have been identified and, unlike us, they excluded pre-print and old VD data articles. Compared to Kazemi et al. [94], we conducted more subgroup and sensitivity analyzes to investigate the heterogeneity between the studies, presenting stratified analyses taking into account study designs and types of analyses. Furthermore, our work is more updated and includes recent studies that were not available when Kazemi et al. [94] conducted the analyses. When we analyzed the association between VD supplementation and type of severity, the estimate in the subgroup of studies related to the ICU was not significant, although there was no significant difference between the various subgroups, in agreement with the meta-analysis conducted by Shah et al. [95]. In general, our meta-analysis is in agreement with those previously published [92, 93, 96–101] even if it is more updated, including the studies up to April 2021, while the previous meta-analyses included the studies up to January 2021 [93].

Furthermore, our meta-analysis was the only one that have investigated the effect of seasonality and these analyses added relevant information because it emerged that a supplementation of VD can probably make sense especially during the winter season, when the frequency of 25OHD deficiencies is greater. Indeed, seasonal analyses revealed a potential greater effect in winter or spring, since the difference in summary estimates between summer versus other seasons was statistically significant. In addition, we have also conducted analyses to verify differences by VD doses. Studies that included an initial bolus were similar to the ones without, and studies with regular doses were similar to those with higher doses however and the summary estimates with regular doses remain statistically significant. This suggests that we may not need to consider very high doses in further studies.

Our analysis included a total of 1197 admitted to the ICU or who needed invasive mechanical ventilation or intubation and hospital stay and more than 910 Covid-19 deaths, thus allowing a statistical power that a single trial rarely manages to achieve.

From our study, stronger and more significant summary estimates have mostly emerged from cohort, prospective and adjusted studies. In general, the latter three are more reliable

than case-control, retrospective studies that do not take into account variables that could act as confounders. This makes our results interesting and more trustable. Although the difference in estimation between randomized trials and observational studies was statistically significant in only one scenario, the summary estimate from the two trials included in the analysis confirmed the associations. The trial by Castillo et al. [24] used calcifediol, which rapidly increases VD levels, compared to vitamin D3 used by Murai et al [26]. The latter presented an estimate that did not indicate a reduced risk but the patients included were diagnosed as severe. These disparities must be kept in mind, since VD is probably more effective in less severe patients. Furthermore, both studies have limitations. The first is not double-blind placebo controlled and it could not take into account, as the authors themselves suggest, the possible role of obesity a possible prognostic factor for severity in Covid-19 patients. Murai's study [26], on the other hand, has a very small and heterogeneous sample in terms of coexisting diseases and therapeutic regimens.

## Limitations

There are some limitations of this study to highlight. First, there was significant heterogeneity among studies which was partly resolved by conducting different sensitivity analysis from which significant associations emerged. Second, the fact that serum VD levels, compared to other analysis, are often measured in subjects suspected of being deficient, may have led some authors to include only subjects for whom they had information on 25(OH)D levels, thus possibly introducing a selection bias. Third, during the analysis for the presence of publication bias, in some scenarios the Begg test and Egger's regression did not lead to the same results; this may be a limitation that we tried to overcome through visual inspection of the funnel plot and imputing potential missing studies. After imputing the missing potential studies, our summary estimates did not change and associations remained statistically significant except in the two scenarios where the summary estimates confirm similar associations, but the estimates lost statistical significance and we observed an increased in heterogeneity between studies. Fourth, the majority of the articles analysed presented data from observational studies with potentially many sources of bias and in addition the quality of these is medium-low. The results must be interpreted with caution because factors such as the type of treatment adopted and the availability of care in hospitals could play an important role on the severity and mortality in Covid-19 patients and, furthermore, no randomized clinical trials has yet demonstrated the clinical benefit of VD supplementation on Covid-19 prognosis. We cannot conclude that low levels of VD increase the risk of infection as the SARS-Cov2 infection occurs 7–10 days before admission, in fact 25OHD may be just a marker of health. These results may be just an effect of reverse causation. However, we think that it was important to include also this analysis and summarize the evidence of eight published studies that may represent food for thought. Furthermore, it is true that these factors can affect the outcome but we do not expect them to differentially affect individuals with high or low levels of VD or who are supplemented or not. None of the published studies investigated these aspects. If there was a non-differential bias it would have to push the estimates towards 1.00 and therefore, if this was the case, it would not be the reason for the significance found. It is also important to keep in mind that conducting RCTs on Covid-19 is very difficult due to the economic burden and the current emergency. It is also important to remember that VD is actually used in hospitals, therefore it is interesting to investigate its role, not to draw definitive conclusions, but to investigate heterogeneity and understand in which circumstances it may be more useful. For example, we showed that VD use in winter seems to be more effective, when 25OHD deficiencies are more frequent, thus suggesting a greater effect in the situation of VDD.

Finally, we should also acknowledge the existence of genetic predisposition of lower VD levels. Therefore, one could not definitively exclude that this genetic trait is linked to other risk factors towards the development of Covid-19 and its severity. This should be considered as a possible limitation of the current analysis, which deserves further studies.

Moreover, a Mendelian randomization study [102] found that low VD levels are not casually associated with Covid-19 susceptibility and severity.

## Conclusions

In conclusions, our meta-analysis of both observational and interventional studies suggests that people with Covid-19 taking VD supplementation, may have a reduced risk of both severity and mortality compared to subjects who do not take VD supplementation, although further RCT are necessary to draw a definitive conclusion on the causal link. Our meta-analysis takes into account the most recent studies and, thanks to the numerous analyses and investigations carried out, could encourage the development of hypotheses that will necessarily have to be tested in randomized controlled experiments to better assess and clarify the role of VD with Covid-19.

## Supporting information

**S1 Checklist.**
(DOCX)

**S1 File.**
(DOCX)

## Acknowledgments

Davide Smussi (Medical Oncology, Department of Medical and Surgical Specialties, Radiological Sciences and Public Health University of Brescia, ASST-Spedali Civili, Brescia, Italy), Miriam Crafa (Medical Oncology, Department of Medical and Surgical Specialties, Radiological Sciences and Public Health University of Brescia, ASST-Spedali Civili, Brescia, Italy), Elisa Sala (SC Oncologia Medica, Asst H S Gerardo Monza, 20900 Monza, Italy), Nicola Ferrari (Medical Oncology, Department of Medical and Surgical Specialties, Radiological Sciences and Public Health University of Brescia, ASST-Spedali Civili, Brescia, Italy), Lavinia Ghiani (Department of Experimental Oncology, IEO, European Institute of Oncology IRCCS, Milan, Italy), Andrea Zorloni (SC Oncologia Medica, Asst H S Gerardo Monza, 20900 Monza, Italy).

## Author Contributions

**Conceptualization:** Susanna Chiocca, Patrizia Gnagnarella, Sara Gandini.

**Data curation:** Oriana D'Ecclesiis, Costanza Gavioli, Chiara Martinoli, Aurora Gaeta, Barbara Zerbato.

**Formal analysis:** Oriana D'Ecclesiis, Federica Bellerba, Aurora Gaeta, Barbara Zerbato, Sara Gandini.

**Funding acquisition:** Susanna Chiocca, Paolo Bossi, Diego Cortinovis, Ferdinando Chiaradonna, Sara Gandini.

**Investigation:** Oriana D'Ecclesiis, Chiara Martinoli, Patrizia Gnagnarella, Sara Gandini.

**Methodology:** Oriana D'Ecclesiis, Costanza Gavioli, Chiara Martinoli, Sara Raimondi, Sara Gandini.

**Project administration:** Susanna Chiocca, Sara Gandini.

**Resources:** Susanna Chiocca, Paolo Bossi, Diego Cortinovis, Ferdinando Chiaradonna, Sara Gandini.

**Software:** Oriana D'Ecclesiis.

**Supervision:** Sara Raimondi, Susanna Chiocca, Patrizia Gnagnarella, Sara Gandini.

**Validation:** Sara Raimondi, Stefania Canova, Aurora Gaeta, Barbara Zerbato, Patrizia Gnagnarella.

**Visualization:** Oriana D'Ecclesiis, Federica Bellerba, Sara Gandini.

**Writing – original draft:** Oriana D'Ecclesiis, Chiara Martinoli, Barbara Zerbato, Sara Gandini.

**Writing – review & editing:** Costanza Gavioli, Chiara Martinoli, Sara Raimondi, Susanna Chiocca, Claudia Miccolo, Paolo Bossi, Diego Cortinovis, Ferdinando Chiaradonna, Roberta Palorini, Federica Faciotti, Federica Bellerba, Costantino Jemos, Emanuela Omodeo Salé, Barbara Zerbato, Patrizia Gnagnarella, Sara Gandini.

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
