## [Decision Letter · Decision Letter 0]

20 Dec 2021

PONE-D-21-35679Vitamin D and SARS-CoV2 infection, severity and mortality: a systematic review and meta-analysisPLOS ONE

Dear Dr. Sara Gandini,

Thank you for submitting your manuscript to PLOS ONE. After careful consideration, we feel that it has merit but does not fully meet PLOS ONE’s publication criteria as it currently stands. Therefore, we invite you to submit a revised version of the manuscript that addresses the points raised during the review process.

We look forward to receiving your revised manuscript.

Kind regards,

Surasak Saokaew, PharmD, PhD, BPHCP, FACP

Academic Editor

PLOS ONE

Journal Requirements:

● A clean copy of the edited manuscript (uploaded as the new *manuscript* file).

4. Please include a copy of Table 1 which you refer to in your text on page 4.

6. We note that this manuscript is a systematic review or meta-analysis; our author guidelines therefore require that you use PRISMA guidance to help improve reporting quality of this type of study. Please upload copies of the completed PRISMA checklist as Supporting Information with a file name “PRISMA checklist”

Reviewers' comments:

Reviewer's Responses to Questions

**Comments to the Author**

1. Is the manuscript technically sound, and do the data support the conclusions?

Reviewer #1: No

Reviewer #2: Yes

2. Has the statistical analysis been performed appropriately and rigorously? 

Reviewer #1: No

Reviewer #2: Yes

3. Have the authors made all data underlying the findings in their manuscript fully available?

Reviewer #1: No

Reviewer #2: Yes

4. Is the manuscript presented in an intelligible fashion and written in standard English?

Reviewer #1: Yes

Reviewer #2: Yes

5. Review Comments to the Author

Reviewer #1: PONE-D-21-35679

Thank you for giving me the opportunity to review the article " Vitamin D and SARS-CoV2 infection, severity and mortality: a systematic review and meta-analysis."

Potential readers are interested in the topic, but there are some fundamental methodology to consider. Furthermore, numerous studies have suggested that vitamin D supplementation could be used to prevent SARS-CoV-2 infection and to treat COVID-19 patients. Aside from that, 25(OH)D or Vitamin D supplementation was associated to COVID-19 composite severity and mortality. For example as indicated below,

1) Crafa, Andrea et al. “Influence of 25-hydroxy-cholecalciferol levels on SARS-CoV-2 infection and COVID-19 severity: A systematic review and meta-analysis.” EClinicalMedicine vol. 37 (2021): 100967. doi:10.1016/j.eclinm.2021.100967

2) Kazemi, Asma et al. “Association of Vitamin D Status with SARS-CoV-2 Infection or COVID-19 Severity: A Systematic Review and Meta-analysis.” Advances in nutrition (Bethesda, Md.) vol. 12,5 (2021): 1636-1658. doi:10.1093/advances/nmab012

3) Rawat, Dimple et al. “"Vitamin D supplementation and COVID-19 treatment: A systematic review and meta-analysis".” Diabetes & metabolic syndrome vol. 15,4 (2021): 102189. doi:10.1016/j.dsx.2021.102189

4) Teshome, Amare et al. “The Impact of Vitamin D Level on COVID-19 Infection: Systematic Review and Meta-Analysis.” Frontiers in public health vol. 9 624559. 5 Mar. 2021, doi:10.3389/fpubh.2021.624559

5) Wang, Zhen et al. “Association of vitamin D deficiency with COVID-19 infection severity: Systematic review and meta-analysis.” Clinical endocrinology, 10.1111/cen.14540. 23 Jun. 2021, doi:10.1111/cen.14540

6) Borsche L, Glauner B, von Mendel J. COVID-19 Mortality Risk Correlates Inversely with Vitamin D3 Status, and a Mortality Rate Close to Zero Could Theoretically Be Achieved at 50 ng/mL 25(OH)D3: Results of a Systematic Review and Meta-Analysis. Nutrients. 2021; 13(10):3596. https://doi.org/10.3390/nu13103596

7) Margarucci, L. M., E. Montanari, G. Gianfranceschi, C. Caprara, F. Valeriani, A. Piccolella, V. Lombardi, E. Scaramucci, and V. Romano Spica. “The Role of Vitamin D in Prevention OF COVID-19 and Its Severity: an Umbrella Review: Vitamin D Versus COVID-19: An Umbrella Review”. Acta Biomedica Atenei Parmensis, vol. 92, no. S6, Oct. 2021, p. e2021451, doi:10.23750/abm.v92iS6.12216.

Whereas, the current evidence suggested that vitamin D deficiency or insufficiency was not significantly linked to susceptibility to COVID-19 infection or its associated death. Vitamin D supplements did not significantly improve clinical outcomes in patients with COVID-19. The overall GRADE evidence quality was low, they suggest that vitamin D supplementation was not recommended for patients with COVID-19.

1) Chen, J., Mei, K., Xie, L. et al. Low vitamin D levels do not aggravate COVID-19 risk or death, and vitamin D supplementation does not improve outcomes in hospitalized patients with COVID-19: a meta-analysis and GRADE assessment of cohort studies and RCTs. Nutr J 20, 89 (2021). https://doi.org/10.1186/s12937-021-00744-y

It is important to note that the emergence and severity of Covid-19 should be regarded as a potential current analysis that needs further investigation, including an Umbrella or Network Meta-Analysis.

Overall, this manuscript is poorly designed, and the work presented probably lacks originality, credibility, and impact. As a result, the authors must revise several points before acceptance.

The major comments are listed below.

Abstract:

1. The results from the analysis cannot provide the answer to the research background/question. The authors also should think about the PICO criteria.

2. The authors stated “We summarised data from 38 studies, including two RCT and 27 cohort-studies: 205565 patients with information on 25OHD status and 2022 taking vitamin D supplementation with a total of 1197 admitted to the ICU or who needed invasive mechanical ventilation or intubation and hospital stay, and more than 910 Covid-19 deaths”, However, this data was not correlated with the results.

Materials and Methods:

Sources of information and search strategies

1. "Vitamin D" may include the MeSH terms "25-OH-D" or "cholecalciferol" or "25-hydroxycholecalciferol" or "calcitriol" or "25-hydroxyvitamin D" or "hydroxycholecalciferols" or "25-hydroxyvitamin D3."

Eligibility criteria

2. What is the main selection criteria in included studies (PICO)? Please define "Low and High 25OHD" as well.

3. The authors should mention about the primary and secondary outcomes of this study (meta-analysis).

Results

Characteristics of eligible studies

1. Please explain Figure 1, and the reasons why studies were excluded from the analysis.

2. The authors did not show the main characteristics of the 38 selected studies. As a result, the reader cannot trust your meta-analysis data, even though the authors show the main characteristics and results of the selected studies in supplementary tables 7, 8, and 9, but the data is incorrect.

3. The potential reader cannot understand why the author select only 8 studied for analyzed in topic “Association between baseline serum vitamin D levels and risk of SARS-CoV-2 infection”, 16 studied for “Association between baseline serum vitamin D levels and severity of Covid-19”, 19 studies for “Association between vitamin D supplementation and the severity of Covid-19”, and 17 studies for “Association between vitamin D supplementation and mortality in patients with Covid-19”

Reviewer #2: This systematic review and meta-analysis assessed the evidence on SARS-CoV2 infection and Covid-19 in relation to deficiency and supplementation of vitamin D. They found significant associations of vitamin D supplementation with Covid-19, encompassing risks of disease worsening and mortality, especially in seasons characterized by 25OHD deficiency and with not severe patients. This study provides benefits for clinical management.

6. PLOS authors have the option to publish the peer review history of their article (what does this mean?). If published, this will include your full peer review and any attached files.

Reviewer #1: No

Reviewer #2: No

---

## [Author Response · Author response to Decision Letter 0]

14 Mar 2022

We thank the editor and reviewers for their helpful suggestions. Hereafter please find a point-by-point reply to reviewers’ comments. 

We highlighted in the manuscript the main changes made from the previous paper.

Reviewer 1

Thank you for giving me the opportunity to review the article " Vitamin D and SARS-CoV2 infection, severity and mortality: a systematic review and meta-analysis."

Potential readers are interested in the topic, but there are some fundamental methodology to consider. Furthermore, numerous studies have suggested that vitamin D supplementation could be used to prevent SARS-CoV-2 infection and to treat COVID-19 patients. Aside from that, 25(OH)D or Vitamin D supplementation was associated to COVID-19 composite severity and mortality. For example as indicated below,

1) Crafa, Andrea et al. “Influence of 25-hydroxy-cholecalciferol levels on SARS-CoV-2 infection and COVID-19 severity: A systematic review and meta-analysis.” EClinicalMedicine vol. 37 (2021): 100967. doi:10.1016/j.eclinm.2021.100967

2) Kazemi, Asma et al. “Association of Vitamin D Status with SARS-CoV-2 Infection or COVID-19 Severity: A Systematic Review and Meta-analysis.” Advances in nutrition (Bethesda, Md.) vol. 12,5 (2021): 1636-1658. doi:10.1093/advances/nmab012

3) Rawat, Dimple et al. “"Vitamin D supplementation and COVID-19 treatment: A systematic review and meta-analysis".” Diabetes & metabolic syndrome vol. 15,4 (2021): 102189. doi:10.1016/j.dsx.2021.102189

4) Teshome, Amare et al. “The Impact of Vitamin D Level on COVID-19 Infection: Systematic Review and Meta-Analysis.” Frontiers in public health vol. 9 624559. 5 Mar. 2021, doi:10.3389/fpubh.2021.624559

5) Wang, Zhen et al. “Association of vitamin D deficiency with COVID-19 infection severity: Systematic review and meta-analysis.” Clinical endocrinology, 10.1111/cen.14540. 23 Jun. 2021, doi:10.1111/cen.14540

6) Borsche L, Glauner B, von Mendel J. COVID-19 Mortality Risk Correlates Inversely with Vitamin D3 Status, and a Mortality Rate Close to Zero Could Theoretically Be Achieved at 50 ng/mL 25(OH)D3: Results of a Systematic Review and Meta-Analysis. Nutrients. 2021; 13(10):3596. https://doi.org/10.3390/nu13103596

7) Margarucci, L. M., E. Montanari, G. Gianfranceschi, C. Caprara, F. Valeriani, A. Piccolella, V. Lombardi, E. Scaramucci, and V. Romano Spica. “The Role of Vitamin D in Prevention OF COVID-19 and Its Severity: an Umbrella Review: Vitamin D Versus COVID-19: An Umbrella Review”. Acta Biomedica Atenei Parmensis, vol. 92, no. S6, Oct. 2021, p. e2021451, doi:10.23750/abm.v92iS6.12216.

Whereas, the current evidence suggested that vitamin D deficiency or insufficiency was not significantly linked to susceptibility to COVID-19 infection or its associated death. Vitamin D supplements did not significantly improve clinical outcomes in patients with COVID-19. The overall GRADE evidence quality was low, they suggest that vitamin D supplementation was not recommended for patients with COVID-19.

1) Chen, J., Mei, K., Xie, L. et al. Low vitamin D levels do not aggravate COVID-19 risk or death, and vitamin D supplementation does not improve outcomes in hospitalized patients with COVID-19: a meta-analysis and GRADE assessment of cohort studies and RCTs. Nutr J 20, 89 (2021). https://doi.org/10.1186/s12937-021-00744-y

It is important to note that the emergence and severity of Covid-19 should be regarded as a potential current analysis that needs further investigation, including an Umbrella or Network Meta-Analysis.

Overall, this manuscript is poorly designed, and the work presented probably lacks originality, credibility, and impact. As a result, the authors must revise several points before acceptance.

Response: The Reviewer is right that indications on this topic are discordant. For this reason the aim of this systematic review and meta-analysis is mainly to quantify uncertainty and identify sources of between-study heterogeneity, in order to understand when and how vitamin D (VD) may be considered a resource and eventually to answer crucial questions to design proper randomized clinical trials. 

We investigated factors that have never been questioned such as dose and seasonality. In particular, it was found that VD supplementation may likely make sense especially during the winter season, when the frequency of 25OHD deficiencies is higher. Indeed, seasonal analyses revealed a potential greater effect in winter or spring, as the difference in summary estimates between summer and other seasons was statistically significant. In addition, we also conducted analyses to test for differences in VD doses, which has never been done before. Studies that included an initial bolus were similar to those without (p-values for difference are not statistically significant), whereas studies with regular doses were similar to those with higher doses, and summary estimates with regular doses remained statistically significant. This suggests that it may not be necessary to consider very high doses in further studies.

We reported the strengths of our study, such as the high sample size and the numerous analysis carried out to investigate the sources of heterogeneity. We have extensively discussed the differences with other meta-analyses (Kazemi, Teshome, Pereira etc., as also indicated by the Reviewer). We would like also to point out that this meta-analysis summarizes the evidence of all the published studies and it allows reaching a high statistical power that single trials cannot reach due to the high costs and long times, especially during a pandemic. 

Network meta-analyses are not necessary at this stage because we could extract all the necessary estimates from the published studies. Umbrella meta-analyses do not allow a deep investigation on sources of bias and variability that are key questions to understand the relationship between vitamin D and Sars-cov2.

The major comments are listed below.

Abstract:

1. The results from the analysis cannot provide the answer to the research background/question. The authors also should think about the PICO criteria.

Response: We agree with the Reviewer, in fact we had included the PICO criteria in the supplemental materials and in addition, we have better defined the objectives of the study in the abstract, as suggested by the Reviewer.

 25OHD and Infection 25OHD and Covid-19

severity Vitamin D supplementation and Covid-19 severity 25OHD and Covid-19 Mortality Vitamin D supplementation and Covid-19

Mortality 

Populations Healthy subjects SARS-CoV2 positive subjects SARS-CoV2 positive subjects SARS-CoV2 positive subjects SARS-CoV2 positive subjects

Exposure/

Intervention Low 25OHD Low 25OHD Current vitamin D supplementation, any dose Low 25OHD Current vitamin D supplementation, any dose

Comparison High 25OHD High 25OHD No current vitamin D supplementation High 25OHD No current vitamin D supplementation

Outcome SARS-CoV2 infection -Admission to hospital ICU

-Need for invasive mechanical ventilation;

-Intubation and hospital length of stay -Admission to hospital ICU

-Need for invasive mechanical ventilation;

- Intubation and hospital length of stay Covid-19 mortality Covid-19

mortality

 Table S2. PICO/PECO strategy

ICU: Intensive Care Unit. 

2. The authors stated “We summarised data from 38 studies, including two RCT and 27 cohort-studies: 205565 patients with information on 25OHD status and 2022 taking vitamin D supplementation with a total of 1197 admitted to the ICU or who needed invasive mechanical ventilation or intubation and hospital stay, and more than 910 Covid-19 deaths”, However, this data was not correlated with the results.

Response: We thank the Reviewer for outlying this point. We have now better clarified how the 38 studies are analysed. In the abstract we specified the following: “We summarised data from 38 eligible studies for at least one outcome, including two RCT and 27 cohort-studies: 205565 patients with information on 25OHD status and 2022 taking vitamin D supplementation with a total of 1197 admitted to the ICU or who needed invasive mechanical ventilation or intubation and hospital stay, and more than 910 Covid-19 death.”

To provide greater clarity we have also modified the text of the results section: "Characteristics of eligible studies" section and the flow chart (Figure1). 

“After application of the inclusion and exclusion criteria (Figure 1), there were 38 independent studies included in the review that were eligible for at least one of the outcome and exposure investigated”. Studies included in the analysis present estimates for multiple outcomes and exposures. In particular:

- Infection scenario: 8 studies included;

- Baseline serum vitamin D levels and severity of Covid-19 scenario: 16 studies included;

- Vitamin D supplementation and the severity of Covid-19 scenario: 6 studies included;

- Baseline serum vitamin D levels and mortality in patients with Covid-19 scenario: 19 studies included;

- Vitamin D supplementation and mortality in patients with Covid-19 scenario: 7 studies included.

Materials and Methods:

Sources of information and search strategies

1. "Vitamin D" may include the MeSH terms "25-OH-D" or "cholecalciferol" or "25-hydroxycholecalciferol" or "calcitriol" or "25-hydroxyvitamin D" or "hydroxycholecalciferols" or "25-hydroxyvitamin D3."

Response: We agree with the Reviewer, the literature search had been conducted with these MeSH terms but for brevity we had reported only "Vitamin D" in the manuscript.

We have updated in the manuscript correctly: “The publications were retrieved using search terms and text words: “Vitamin D” or "25-OH-D" or “25-hydroxycholecalciferol” or “25-hydroxyvitamin D” or “25-hydroxyvitamin D3” or “cholecalciferol” or “calcitriol” or “hydroxycholecalciferols” in combination with “Covid” or “SARS-CoV2” or “Covid-19” or “Covid-19 mortality” or “Covid-19 death or “Covid-19 severity” or “SARS-CoV2 infection” without any restriction. The database search was supplemented by consulting the bibliography of the articles, reviews and published meta-analysis.”

Eligibility criteria

2. What is the main selection criteria in included studies (PICO)? Please define "Low and High 25OHD" as well.

Response: We agree with the Reviewer, in fact we had included the PICO criteria in the supplemental materials.

Thanking the Reviewer for the important suggestion, we defined low and high levels of 25OHD both in the table S7 and S8 and in manuscript section “study outcomes and exposure”.

Thanks to the Reviewer’s suggestion, we have included an additional important analysis to the manuscript. Through dose-response analyses, it was found that different cut-offs of vitamin D levels of studies did not change the risk of infection (p=0.9), mortality (p=0.15), or severity (p=0.66). These findings were incorporated and discussed in the manuscript.

3. The authors should mention about the primary and secondary outcomes of this study (meta-analysis).

Response: We agree with the Reviewer and have better specified primary and secondary outcomes in the manuscript: “We considered two primary outcomes: Covid-19 severity, in terms of need for admission to hospital intensive care unit (ICU) or for invasive mechanical ventilation or intubation and hospital length of stay (1) or mortality (2). In addition, we considered one secondary outcome: SARS-CoV2 infection (3).

We considered two main factors linked to VD exposure: (1) baseline VD deficiency or insufficiency (analysed as lowest versus highest 25(OH)D levels, the most studies used thresholds equal to 20nmol / L or 30 nmol / L), and (2) VD supplementation.”

Results

Characteristics of eligible studies

1. Please explain Figure 1, and the reasons why studies were excluded from the analysis.

Response: We agree with the Reviewer and have specified more in the manuscript: “…35 studies were excluded because: they contained neither risk estimates nor sufficient data for calculation; they did not measure VD levels; they were not consistent with the objectives of our study; they were not peer-reviewed or due to study design.”

2. The authors did not show the main characteristics of the 38 selected studies. As a result, the reader cannot trust your meta-analysis data, even though the authors show the main characteristics and results of the selected studies in supplementary tables 7, 8, and 9, but the data is incorrect.

Response: We have clarified this issue further, as well as mentioned in point 2.

We agree with the Reviewer and have better clarified in the text "Characteristics of eligible studies" section how the 38 studies were analysed. 

3. The potential reader cannot understand why the author select only 8 studied for analyzed in topic “Association between baseline serum vitamin D levels and risk of SARS-CoV-2 infection”, 16 studied for “Association between baseline serum vitamin D levels and severity of Covid-19”, 19 studies for “Association between vitamin D supplementation and the severity of Covid-19”, and 17 studies for “Association between vitamin D supplementation and mortality in patients with Covid-19”

Response: Thank for pointing out that this aspect was not clear. We agree with the auditor and have clarified this issue further in the previous item.

Reviewer 2

This systematic review and meta-analysis assessed the evidence on SARS-CoV2 infection and Covid-19 in relation to deficiency and supplementation of vitamin D. They found significant associations of vitamin D supplementation with Covid-19, encompassing risks of disease worsening and mortality, especially in seasons characterized by 25OHD deficiency and with not severe patients. This study provides benefits for clinical management.

Response: We thank the Reviewer for the kind words. We have brought some novelties through our study so far never analyzed in other similar work, namely that VD supplementation may likely make sense especially during the winter season, when the frequency of 25OHD deficiencies is higher and not be necessary to consider very high doses in further studies. We are glad that this is noticed and appreciated.

---

## [Decision Letter · Decision Letter 1]

29 Apr 2022

Vitamin D and SARS-CoV2 infection, severity and mortality: a systematic review and meta-analysis

PONE-D-21-35679R1

Dear Dr. Sara Gandini,

We’re pleased to inform you that your manuscript has been judged scientifically suitable for publication and will be formally accepted for publication once it meets all outstanding technical requirements.

Kind regards,

Surasak Saokaew, PharmD, RPh, PhD, BPHCP, FACP, FCPA

Academic Editor

PLOS ONE

Additional Editor Comments (optional):

Reviewers' comments:

Reviewer's Responses to Questions

**Comments to the Author**

1. If the authors have adequately addressed your comments raised in a previous round of review and you feel that this manuscript is now acceptable for publication, you may indicate that here to bypass the “Comments to the Author” section, enter your conflict of interest statement in the “Confidential to Editor” section, and submit your "Accept" recommendation.

Reviewer #1: All comments have been addressed

Reviewer #2: (No Response)

2. Is the manuscript technically sound, and do the data support the conclusions?

Reviewer #1: Yes

Reviewer #2: Yes

3. Has the statistical analysis been performed appropriately and rigorously? 

Reviewer #1: Yes

Reviewer #2: Yes

4. Have the authors made all data underlying the findings in their manuscript fully available?

Reviewer #1: Yes

Reviewer #2: Yes

5. Is the manuscript presented in an intelligible fashion and written in standard English?

Reviewer #1: Yes

Reviewer #2: Yes

6. Review Comments to the Author

Reviewer #1: Thank you for revising the article "Vitamin D and SARSCoV2 infection, severity and mortality: a systematic review and meta-analysis." The authors have adequately addressed all comments raised in the previous round of review. I recommend accepting this manuscript to be published in PLOS ONE.

Reviewer #2: It is a nice manuscript the present a meta-analysis about the effect of vitamin-D on mortality and severity of SARS-CoV2 infection. The study is well presented and cover an interesting topic.

7. PLOS authors have the option to publish the peer review history of their article (what does this mean?). If published, this will include your full peer review and any attached files.

Reviewer #1: No

Reviewer #2: **Yes: **Asst. Prof. Sukrit Kanchanasurakit

---

## [Editor Report · Acceptance letter]

7 Jun 2022

PONE-D-21-35679R1 

Vitamin D and SARS-CoV2 infection, severity and mortality: a systematic review and meta-analysis 

Dear Dr. Gandini:

I'm pleased to inform you that your manuscript has been deemed suitable for publication in PLOS ONE. Congratulations! Your manuscript is now with our production department. 

Kind regards, 

on behalf of

Dr. Surasak Saokaew 

Academic Editor

PLOS ONE